

# Atmosphere Density Measurements Using GPS Data from Rigid Falling Spheres

Yunxia Yuan[1], Nickolay Ivchenko[1], Gunnar Tibert[2], Marin Stanev[3], Jonas Hedin[3], and Jörg Gumbel[3]

[1]School of Electrical Engineering, Royal Institute of Technology KTH, Stockholm, Sweden
[2]School of Engineering Sciences, Royal Institute of Technology KTH, Stockholm, Sweden
[3]Department of Meteorology, Stockholm University, Stockholm, Sweden

*Correspondence to:* Nickolay Ivchenko (Email:nickolay@kth.se)

**Abstract.** Atmospheric density profiles in the stratosphere and mesosphere are determined by means of low cost Global Positioning System (GPS) receivers on in situ rigid falling spheres released from a sounding rocket. Values below an altitude of 80 km are obtained. Aerodynamic drag relates atmospheric densities to other variables such as velocities of spheres, drag coefficients, and reference area. The densities are reconstructed by iterative solution. The calculated density is reasonably accurate, with deviation within 10% with respect to the European Centre for Medium-range Weather Forecasts ( ECMWF) reference value. The atmospheric temperature and wind profiles are obtained as well, and compared to independent data.

## 1  Introduction

The middle atmosphere, comprising altitudes 10–100 km, constitutes an important part of the Earth's climate system. A wide range of dynamical, radiative and chemical interactions connect local conditions to processes in the lower and upper atmosphere. Remote sensing techniques from the ground and from space provide us with comprehensive information about the state and variability of the middle atmosphere. However, in many cases in situ measurements are needed to gain a detailed understanding of the processes that control local conditions and phenomena. In particular, there is a need to characterize background atmospheric conditions in terms of temperature, density and winds, either as stand-alone studies or as a complement to in situ measurements of other specific parameters.

In situ falling sphere experiments launch spheric probes to middle and upper atmosphere by sounding rockets, in order to study the atmosphere at high altitudes. They can be classified into passive falling spheres and active falling spheres (Li et al., 2013). The passive falling spheres do not carry sensors to take measurements, whereas the active falling spheres take measurements themselves. Since the 1950s, falling sphere experiments to study the atmosphere have been performed to derive middle and upper atmospheric parameters. This method has been updated due to development of new technologies, for instance, from transponder and radar (passive falling spheres) to accelerometers (active falling spheres), and from big inflatable spheres to small rigid spheres. Bartman et al. (1956) derived densities and temperatures by using an inflatable sphere with transponder and antenna. Otterman et al. (1961) proposed to observe atmospheric densities and wind velocities by using a falling inflatable sphere with accelerometers. Faucher et al. (1963) and Faucher and Morrissey (1971) retrieved densities at different altitudes by using an inflatable sphere instrumented with accelerometers. Salah (1967) determined densities and temperatures at strato-





mesospheric altitudes, using rigid spheres re-entering the atmosphere, whose trajectory parameters were measured by radar. Schmidlin et al. (1991) stated that even though the densities from inflatable falling spheres contain some linear bias that is indistinguishable from the measurement error, the linear bias does not significantly influence the temperatures derived from the densities.

Otterman et al. (1961) used a falling sphere to measure wind velocities, whereas other sources (Faucher and Morrissey, 1971; 30 Faucher et al., 1963; Salah, 1967) assumed that wind velocities were negligible when calculating densities and temperatures. Martineau (2012) employed rigid falling spheres to measure wind speeds based on the rotational and the translational motion of the spheres. The instrumentation and the mathematical model to derive the density were presented, however, the details relating to the wind speed calculations and corresponding results were not shown. An experiment using GPS raw data logged inside a rigid sphere (Bordogna et al., 2013) was developed in the frame of the Rocket EXperiment for University Students (REXUS) 35 programme, (REXUS, 2017). This experiment was further employed in a research sounding rocket launched in 2016, as we report here.

The inflatable falling sphere (Schmidlin et al., 1991) has been extensively used for retrieving profiles of atmospheric parameters. Lübken et al. (1994) have made an extensive comparison between various methods of determining densities and temperatures, finding excellent agreement of the falling sphere data. Campaigns with multiple inflatable sphere flights have 40 been carried out in Northern Norway (Schmidlin and Schauer, 2001), on Svalbard (Lübken and Müllemann, 2003), Antarctics (Lübken et al., 2004). The availability of the technique has diminished as the small rockets used have become discontinued.

This paper investigates determination of the atmospheric density, wind and temperature profiles using GPS data from rigid spheres flown on a sounding rocket. First, we present the GPS data processed to derive aerodynamic accelerations of the spheres. We then proceed to summarize the approximations for drag coefficients for different altitude ranges, since drag co- 45 efficients are a key to compute densities. Then, based on aerodynamics, the densities are obtained by iterating a dynamical equation. Using the hydrostatic equilibrium relation, the temperature profiles can be retrieved. When the aerodynamic accelerations only consist of drag components, wind velocities can be derived easily. We conclude with a comparison of the results with other observations during the period, and a discussion of errors.

## 2   Data

This paper makes use of GPS data from the Local Excitation and Effects of Waves on Atmospheric VErtical Structure (LEE-WAVES) falling rigid spheres to calculate densities and temperatures, and compares the derived results to the values from the NRLMSISE-00 model (Picone et al., 2002), radiosonde measurements, Lidar, as well as ECMWF analysis (ECMWF, 2015).

### 2.1   LEEWAVES sounding rocket experiment

LEEWAVES was a Swedish complement to the international multi-instrument campaign GW-LCYCLE targeting the propaga- 55 tion of atmospheric gravity waves in high latitude middle atmosphere. The campaing included a sounding rocket experiment to



characterize atmospheric properties using a rigid falling sphere. The rocket was launched from Esrange Space Centre, Sweden at 21:09 UT on Feb. 2, 2016.

The LEEWAVES rocket experiment consisted of four spherical free flying units (FFUs), based on the design from the previous demonstration in the REXUS experiment (Bordogna et al., 2013). Each unit had a mass of 0.413 kg, a diameter of 124 mm, and carried a GPS data logger, which recorded raw L1 GPS signal downconverted to 2 MHz with 2 bit resolution. The data were recorded from the ejection from the sounding rocket at around 80 km. An apogee of 138 km was reached. The FFUs landed with parachutes, were recovered and the data were read out. Afterwards, GPS trajectory solutions including positions and velocities were obtained using a GPS software receiver (Borre et al., 2007), following the procedure detailed in (Yuan et al., submitted to Journal of Geodetic Science). A global optimisation method, using both pseudorange and Dopper frequency observables is applied to produce a trajectory solution at 10 ms temporal resolution. Here we present data from two FFUs with the most complete coverage, further on denoted as LW2 and LW4.

The direct outputs from GPS data processing are positions and velocities of a FFU, which are expressed in the Earth Centered Earth Fixed (ECEF) frame. Figure 1 show the altitudes, velocities and accelerations of LW2 and LW4 in the local frame of East North Up (ENU).

The ECEF frame has the origin at the center of the Earth, the Z axis points to the north pole of the Earth, the X axis points towards the intersection point of the Greenwich meridian and the equator, the Y axis forms the right hand coordinate frame together with X axis and Z axis. As the name suggests, this frame is fixed to the Earth and rotates with the Earth. The accelerations were first calculated in the ECEF frame and were then transformed into the ENU frame.

The ENU frame has the origin at the center of mass of a FFU, the X axis is towards the east in the horizontal plane, the Y axis points to the north in the horizontal plane, the Z axis points upwards and forms the right hand coordinate system.

Three regions of the trajectory can be distinguished according to variations of accelerations, 1. The first is "free fall region" starting at the sphere ejection and until 240 s, corresponding to about 60 km on the downleg. In this interval the gravity dominates over the aerodynamic drag, resulting in almost linear change in velocity. The second region is the deceleration, between 240 s and 280 s, corresponding to altitudes between about 60 km and 25 km on the downleg. Here the air drag peaks, resulting in a vertical acceleration of over 4 g as the fall velocity decreases to below $100\,\mathrm{ms^{-1}}$. The final interval is after 280 s, where the drag forces roughly balance the gravity force, and the spheres fall with their terminal velocity, dependent on local density.

Note that these positions and velocities retrieved from the GPS data analysis refer to the phase center of the receiver antenna on the FFU, instead of the center of mass, as there is a small distance between the antenna and the mass center. Due to wobbling of the FFU, the antenna phase center traces out a complicated curve with respect to the center of mass, resulting in oscillating velocities. We apply a low-pass digital filter in Matlab to approximate the velocities of the mass center. The low-pass filter is 5th-order Butterworth filter with normalized cutoff frequency 0.1 Hz. The accelerations $a$ of the mass center are found by numerical differentiation, shown in Fig. 1(c). The filter parameter selection is a trade-off between residuals and frequencies allowed to pass. The larger the cutoff frequency is, the smaller the residuals are, yet the higher components the passing frequencies have. Figure 2 shows velocities, aerodynamic accelerations (see Sect. 3.4), angles between the velocity and





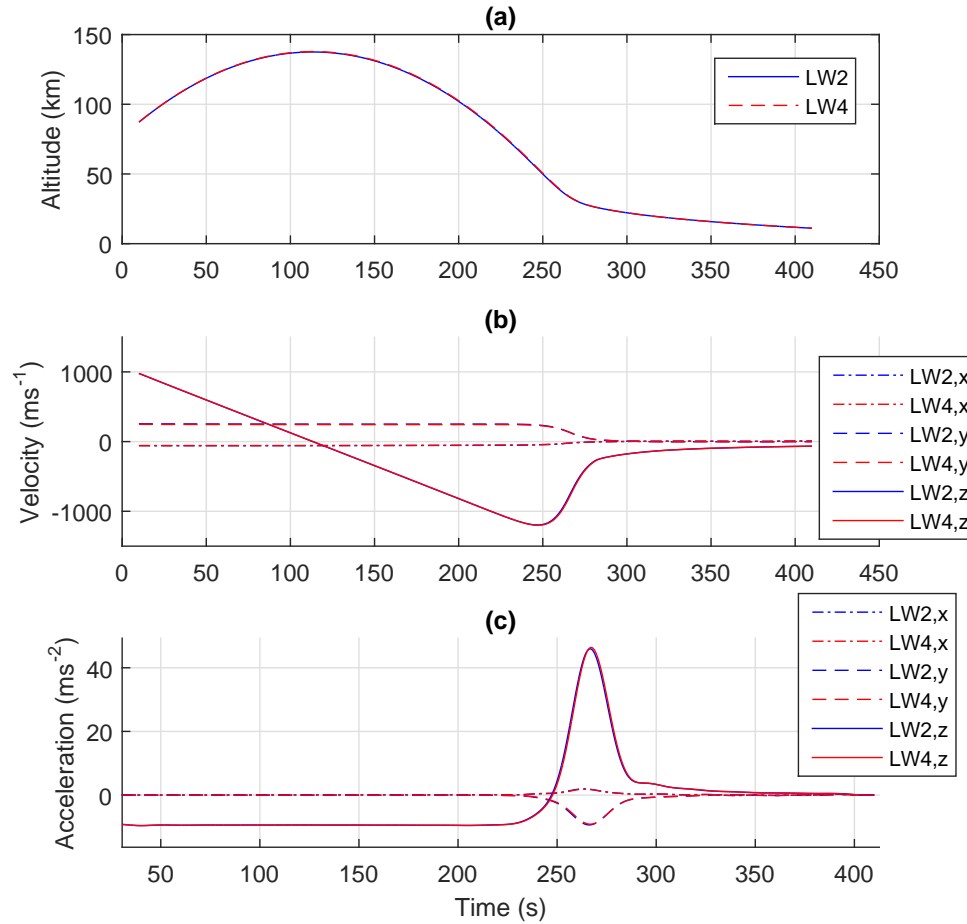

**Figure 1.** Altitudes, velocities and accelerations of FFUs in local ENU frame, LWX is denoting FFU X





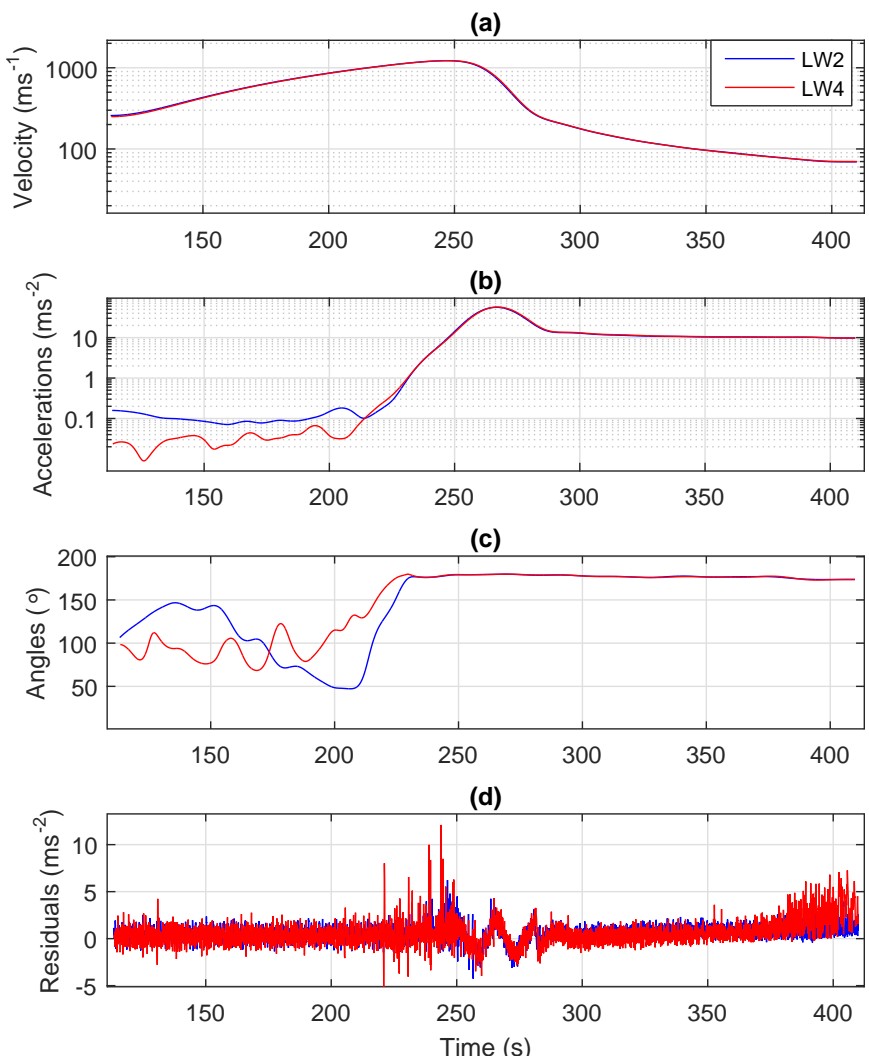

**Figure 2.** Aerodynamic acceleration information for LW2 and LW4



the aerodynamic acceleration vectors, and filtering residuals of accelerations. From Fig. 2(b), above 80 km, the accelerations are very small, around 0.1 $\mathrm{ms}^{-2}$ for LW2, and even smaller for LW4. The aerodynamic acceleration is about 0.3 $\mathrm{ms}^{-2}$, at 224 s, corresponding to 80 km approximately, increasing to around 1 $\mathrm{ms}^{-2}$, at 231 s, corresponding to 72 km approximately. Below 72 km, the accelerations become larger than 1 $\mathrm{ms}^{-2}$. Below 54 km, i.e. after 247 s, they reach around 10 $\mathrm{ms}^{-2}$ or

larger. From Fig. 2(d), we can see that the filtering residuals are much larger than the filtered accelerations before 224 s, which imply that the uncertainties of these accelerations are large. In Fig. 2(c), the angles for LW4 become larger than 173° after 224 s, which those for LW2 are larger than 150° after 224 s, larger than 170° after 228 s. Angles close to 180° mean that the aerodynamic accelerations are close to opposite to the velocities, as expected for air drag. Both Fig. 2 (b) and (c) imply the consistent conclusion: the acceleration estimates are valid below 70–80 km.

## 2.2 Reference data

For comparison with the falling sphere data, we also use data from models and other in situ observations obtained in the LEEWAVES campaign.

NRLMSISE-00 is an empirical global model that models densities and temperatures of the Earth's atmosphere from ground to space.

Radiosonde measurements provide density, temperature and winds of the atmosphere from the ground to the middle strato-sphere, with top altitudes of typically 25–35 km. Measurements are based on balloon-borne VAISALA radiosondes RS92-SGP, (VAISALA, 2013). Typical accuracies of the temperature measurements are 0.2 K (Nash et al., 2010). Note, however, that these balloon measurements do not provide real vertical profiles above Esrange but drift with altitude in the predominantly eastward wind field.

The temperature comparison data has been collected using the Esrange Rayleigh / Mie / Raman lidar instrument stationed at the launch site of the LEEWAVES rocket (Blum and Fricke, 2005). For this study, only the most sensitive Rayleigh channel with a height range between 30 km and 70 km in the aerosol-free part of the atmosphere has been used in the comparison with the LEEWAVES probe temperature results. Because of tropospheric cloudiness, lidar information was unfortunately not available concurrent with the LEEWAVES launch. Instead, comparisons in this paper are based on lidar measurements obtained during

the day prior to the launch. Error estimates are based on the propagation of the error from the integration of the hydrostatical equation and the statistical error in count rates at different altitudes.

The meteorological density and temperature information is based on the operational analyses of the Integrated Forecast System (IFS) of ECMWF. We use data from the global deterministic High-RESolution (HRES) IFS cycle 41r1 that became operational on the May 12, 2015 and provides a 16 km horizontal resolution and 137 model level with a model top at 0.01 hPa.

Figures 3–5 show densities, temperatures, and winds from various data sources.

Lidar data shown are a 10 hour average taken on Feb. 1, 2016 at 13:44–23:44 UT. Radiosonde data include five sets of data, from balloons launched at 16:55 UT on Feb. 1 (R1), 16:30 UT on Feb. 2 (R2), 21:39 UT on Feb. 2 (R3), 00:23 UT on Feb. 3 (R4), and 03:28 UT on Feb. 3 (R5). NRLMSISE-00 generated the data at the launch time of LEEWAVES, at Esrange



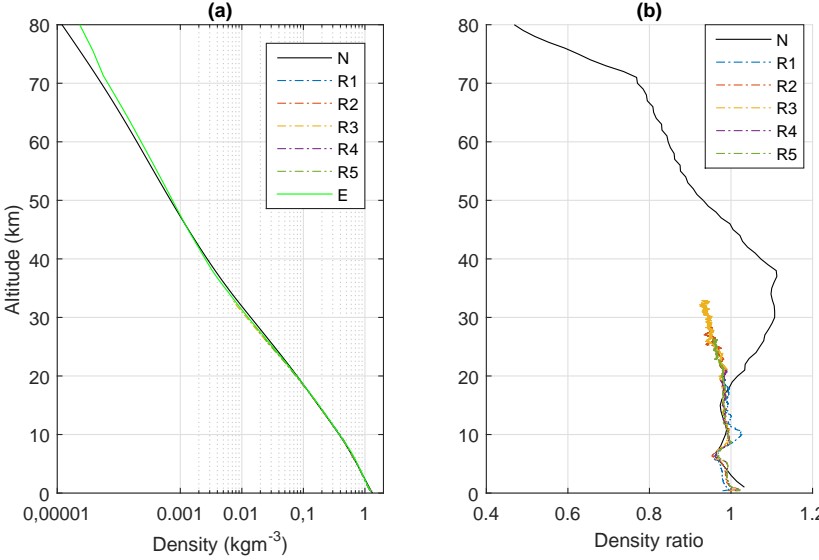

**Figure 3.** Left: Densities from NRLMSISE-00 (N), Radiosonde (Ri, i=1,...,5), and ECMWF (E); Right: Ratio of other densities to the ECMWF density

(21:09 UT, Feb. 2, 2016, 21.083° E, 67.891° N), with the model parameters taken for the actual activity level for that day. The ECMWF data analysis is shown for three locations: at Esrange (21.083° E, 67.891° N), at the apogee (20.884° E, 68.264° N), and at 20 km in the downleg (20.696° E, 68.598° N), at 21:00 on Feb. 2, 2016 UTC, which are close to the flight area and time.

The densities from NRLMSISE-00 are statistically reliable at high altitudes, here, we trust those above 70 km. Whereas the densities from ECMWF are reliable below 50 km, as the altitude goes up, the uncertainties increase (Le Pichon et al., 2015). In Fig. 3, we show NRLMSISE-00 and ECMWF data for 0–80 km to illustrate their discrepancies. Radiosonde provides densities up to 35 km. Below 50 km, other densities are close to the ECMWF densities within 10%. Above 50 km, the NRLMSISE-00 densities deviate from the ECMWF densities gradually, the ratio decreases from 0.94 to 0.52 approximately.

As for temperatures, Lidar provides temperatures from 25 km to 80 km. Radiosondes provide data below around 35 km. ECMWF covers the whole altitude range below 80 km. In general, the three data sets of temperature are in good agreement. Radiosondes and ECMWF have better agreement, with differences smaller than 3 K below 25 km, but slightly larger differences above 25 km, about 10 K. The ECMWF data for different sites have large differences between 50 km and 70 km, the maximum difference reaching 10 K. The Lidar temperatures also have large differences with the ECMWF data from 45 km to 72 km, the maximum difference reaches 20 K.

In Fig. 5, E1 and Ri, i=1,...,5 represent the wind speeds at the launch site, Esrange. They are close to each other, most of the differences are within $20 \text{ ms}^{-1}$, and the absolute values are smaller than $20 \text{ ms}^{-1}$. E2 and E3 are the wind speeds at apogee and 20 km in the downleg. They match well, but are larger than $20 \text{ ms}^{-1}$ between 13 km and 73 km for the E/W direction,





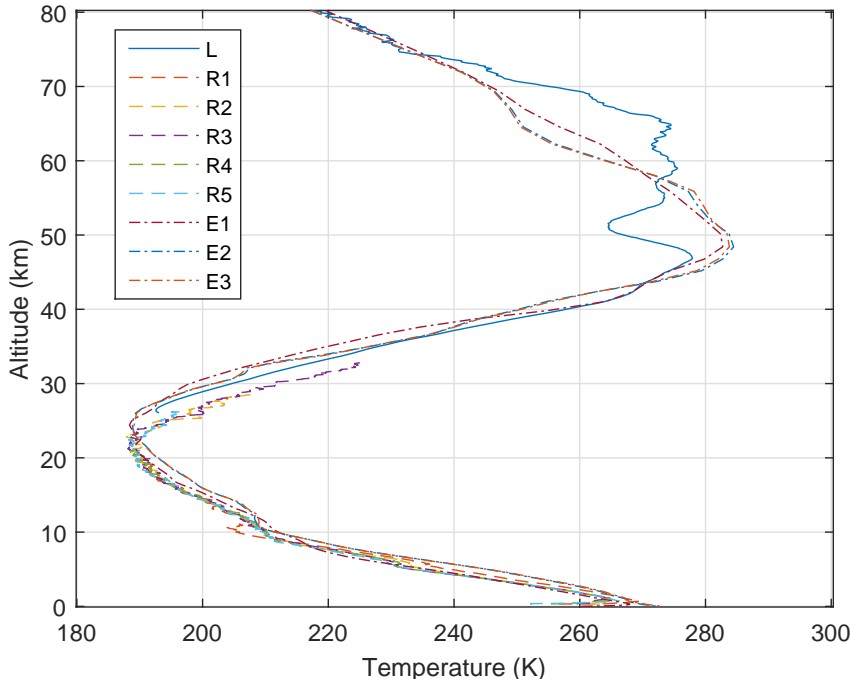

**Figure 4.** Temperatures from Lidar (L), Radiosonde (Ri, i=1,...,5) and ECMWF (Ei, i=1,...,3)

between 35 km and 70 km for the N/S direction. The maximum value is 63 ms$^{-1}$ for the E/W direction, 57 ms$^{-1}$ for the N/S direction.

## 2.3 Meteorological conditions

In general the northern hemisphere winter season 2015/2016 is characterized by record high temperatures near the surface and record cold temperatures in the stratosphere. The unusual conditions in the middle atmosphere in early 2016 are described in (Matthias et al., 2016). We describe the conditions for troposphere and stratosphere for the season, and here focus on the launch day.

### 2.3.1 Troposphere

The winter season 2015/2016 has been characterized as the warmest arctic winter on record (Cullather et al., 2016). While northern Scandinavia did not experience temperature anomalies, large parts of the Arctic and the Bering sea experienced temperature anomalies up to 6 K in Jan.–Mar. 2016 average compared to the baseline period 1981–2016 (Overland et al., 2016).





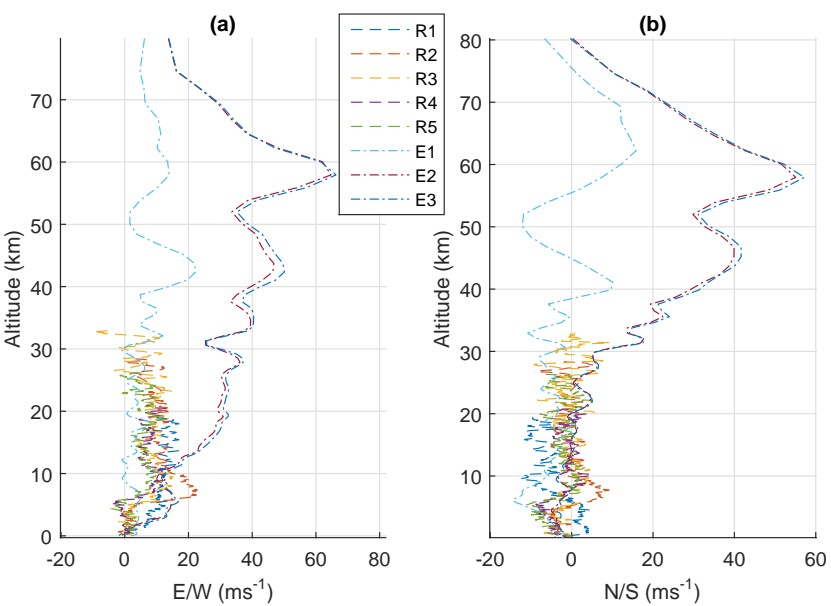

**Figure 5.** Winds from Radiosonde (Ri, i=1,...,5) and ECMWF (Ei, i=1,...,3)

### 2.3.2 Stratosphere

Contrary to the tropospheric conditions, we observe record cold temperatures in the Arctic stratosphere of winter 2015/2016. A 155
strong and stable stratospheric polar vortex persisted until the final major warming (Manney and Lawrence, 2016) on Mar. 5–6
2016. A strong meridional temperature gradient led to adverse conditions for the propagation of planetary waves and therefore
led to the confinement and subsequent continuous cooling of stratospheric air below 185 K from late Dec. 2015 until the end
of Jan. 2016 (Voigt et al., submitted). Comparing these winter observations to the ERA-INTERIM reanalysis dataset, we find
that the low Arctic temperatures in January with a minimum temperature of 179 K broke the minimum temperature record for 160
meteorological data since 1979 (Manney and Lawrence, 2016).

### 2.3.3 Launch day

Figure 6 shows the meteorological condition on the launch day. The launch of the LEEWAVES rocket took place, directly
inside the stratospheric polar vortex at the end of its coldest period. Potential vorticity on isentropic surfaces is conserved and
regions of high potential vorticity at the Earth poles have been shown as stable indicators of the polar vortex for weeks to 165
months (Hoskins et al., 1985). At the potential temperature surface 675 K corresponding with approximate 27 km altitude, we
still see an elongated polar vortex with a southern edge in the potential vorticity around northern Germany and a temperature
minimum above Sweden.





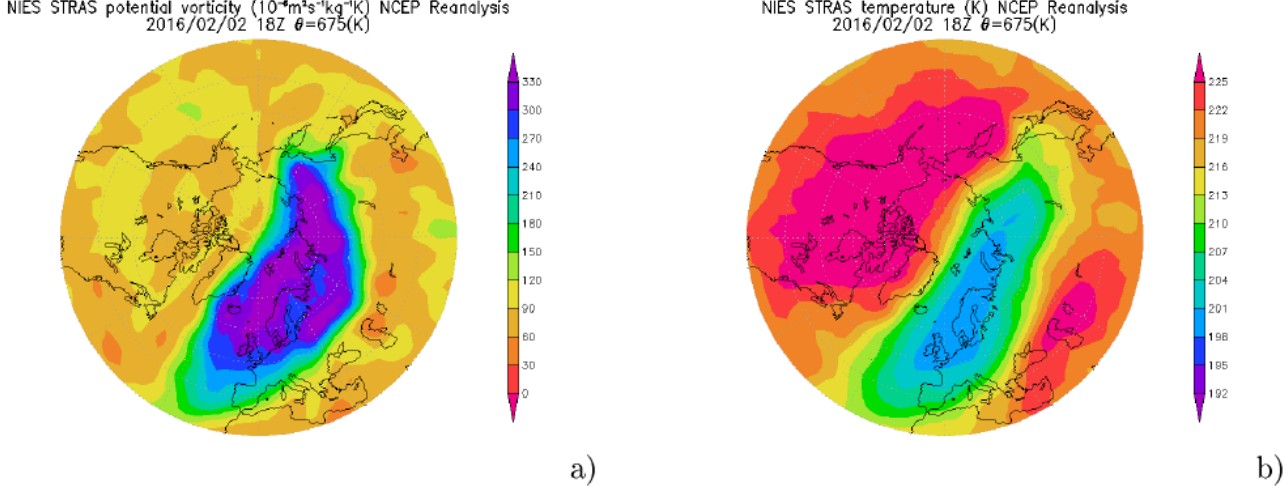

**Figure 6.** a). Stratospheric potential vorticity on the 675 K (27 km) isentropic surface as a proxy for the polar vortex on the Feb. 2, 2016 18:00 UT b). Stratospheric temperature on the Feb. 2, 2016 18:00 UT. Figures from the Center for Global Environmental Research (CGER) using meteorological data provided by the National Center for Environmental Prediction (NCEP)

## 3 Analysis

The analysis principle of the falling sphere is based on the aerodynamic drag expression, which relates the drag acceleration to the velocity.

### 3.1   Aerodynamic force

For a rotating FFU, the aerodynamic force is composed of a drag force opposite to its velocity direction and a lift force perpendicular to the velocity, induced by the Magnus effect, i.e., Magnus lift (Martineau, 2012; Seifert, 2012; Volkov, 2009).

The drag force vector $\boldsymbol{D}$ acting on a FFU can be expressed as

$$\boldsymbol{D} = \frac{1}{2}\rho V^2 A C_D \frac{-\boldsymbol{V}}{V} = -\frac{1}{2}\rho A C_D V \boldsymbol{V} \tag{1}$$

where $\rho$ is the density of the atmosphere, $\boldsymbol{V}$ is the velocity vector of the FFU with respect to the atmosphere, $V$ is the magnitude, $A$ is the reference area of the FFU (maximum cross section area for a sphere), and $C_D$ is the drag coefficient. Considering that horizontal winds exist,

$\boldsymbol{V} = \boldsymbol{v} - \boldsymbol{w}$                                               (2)

where $\boldsymbol{v}$ denotes the velocity vector of the mass center of the FFU with respect to the surface of the Earth, $\boldsymbol{w}$ denotes the wind velocity vector.


The Magnus effect is the basic flow phenomenon responsible for the sideways deviation of a sphere rotating around an axis perpendicular to the flight direction from its initial straight path. The Magnus lift vector $\boldsymbol{L}$ (Volkov, 2009) is

$$\boldsymbol{L} = \frac{1}{2}\rho A R_s C_L \boldsymbol{\omega} \times \boldsymbol{V} \tag{3}$$

where $R_s$ is the radius of the FFU, $C_L$ is the lift coefficient, $\boldsymbol{\omega}$ is the angular velocity vector of the sphere. Generally, the lift coefficient is smaller than 1.5 according to Volkov (2009). From the authors' preliminary data analysis on angular velocities of the FFUs, the angular velocity is around 5 rad/s, the ratio between the lift and the drag forces is around $R_s C_L \omega/(C_D V)$ $\approx 0.06 \times 5/300 = 0.001$, $C_L$ and $C_D$ being the same order of magnitude. Hence, the Magnus lift force can be neglected in the subsequent analysis.

While the Magnus force can make the dynamic model very precise, it requires attitude determination, making the model more complex.

## 3.2 Drag coefficient

The drag coefficient depends on the Reynolds and Mach numbers (Bartman et al., 1956).

The Reynolds number gives a measure of the ratio between the inertial and viscous forces of the flow,

$$Re = \frac{\rho V L}{\mu} \tag{4}$$

where $L$ is a reference length and $\mu$ is the dynamic viscosity of the flow medium. For a sphere, the diameter is used as the reference length. The dynamic viscosity depends on the temperature; for air it is given by Sutherland's law (Anderson Jr, 2007)

$$\frac{\mu}{\mu_{ref}} = \left(\frac{T}{T_{ref}}\right)^{3/2} \frac{T_{ref}+110}{T+110} \tag{5}$$

where $\mu_{ref}$ is a reference dynamic viscosity at a reference temperature $T_{ref}$. $T$ is the absolute temperature. Here, $\mu_{ref} = 1.7894 \times 10^{-5}$ kgs$^{-1}$m$^{-1}$ and $T_{ref} = 288.16$ K were used.

The Mach number $Ma$ is defined as

$$Ma = \frac{V}{s} = \frac{V}{\sqrt{\gamma R_g T}} \tag{6}$$

where $s$ is the speed of sound, $\gamma$ is the ratio of specific heat, $R_g$ is the specific gas constant ($R_g = 287.04$ J kg$^{-1}$ K for dry air). For a large temperature range, the ratio of specific heat $\gamma$ should be computed as (Sutherland and Bass, 2004)

$$\gamma = A_0 + A_1 T + A_2 T^2 + A_3 T^3 + A_4 T^4 + A_5 T^5 \tag{7}$$

where $A_0$=1.371, $A_1$=2.460×10$^{-4}$, $A_2$=-6.436×10$^{-7}$, $A_3$= 5.200×10$^{-10}$, $A_4$=-1.796×10$^{-13}$, and $A_5$=2.182×10$^{-17}$.

Another important parameter is Knudsen number that is a characteristic value for gas dynamics (O'Hanlon, 2004),

$$Kn = \frac{Ma}{Re}\sqrt{\frac{\gamma\pi}{2}} \tag{8}$$



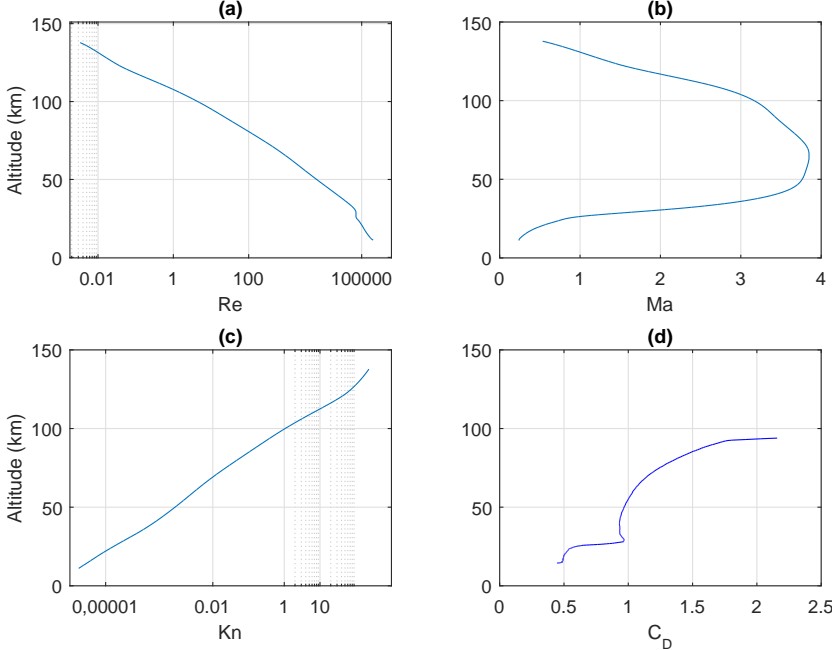

**Figure 7.** Reynolds, Mach, Knudsen numbers and drag coefficients as functions of altitude

According to Knudsen number, the gas dynamics is classified into continuum flow ($Kn < 0.01$), transitional flow ($0.01 \leq Kn < 1$), and free molecular flow ($Kn \geq 1$) (O'Hanlon, 2004).

Figure 7 presents Reynolds numbers, Mach numbers, and Knudsen numbers for the downleg trajectory, as well as drag coefficients below 92 km. These are the nominal values according to densities and temperatures from NRLMSISE-00, as it can provide data up to 150 km. The general corresponding relation between altitudes and flow regimes is, below 70 km, continuum flow, $Kn < 0.01$; between 70 km and 100 km, transition flow, $0.01 \leq Kn < 1$; above 100 km, free molecular $Kn \geq 1$.

The drag coefficient is related to the density and the temperature through the Mach number and the Reynolds number. We obtain the drag coefficient through scatter interpolating experiment data from wind tunnels and ballistic flights shown in Fig. 8. These experiment data are from Bailey and Hiatt (1972), with estimated uncertainties of $\pm 2\%$. These data can provide drag coefficient data for altitudes up to 95 km and down to 16 km.

## 3.3 Temperature

To obtain Mach and Reynolds numbers, it is necessary to know the temperatures. At the start of the analysis, the model values are used to get initial approximations of the numbers. The density profiles obtained from the inversion of the drag will not be in hydrostatic equilibrium, and a temperature profile corresponding to the observed density profile can be derived.



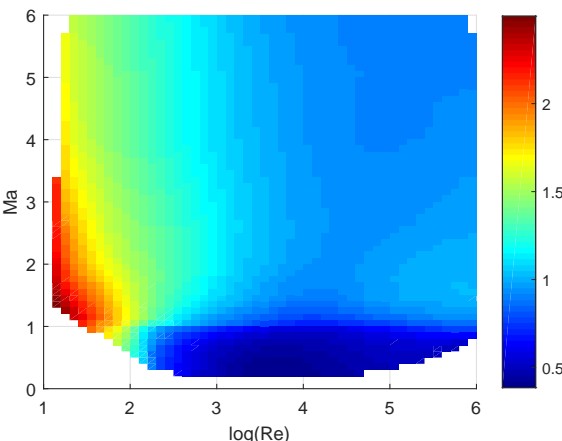

**Figure 8.** Experiment data for drag coefficients (in color), shown as a function of Reynolds number (logarithm with base 10 on x axis) and Mach number (on y axis)

The ideal gas law relates the pressure $p$ to the mass density $\rho$,

$$p = \rho R_g T \qquad (9)$$

The hydrostatic equation describes the pressure balance,

$$dp = -\rho g dz \qquad (10)$$

where $dp$ is a small change in pressure and corresponding to a small change in altitude $dz$.

Integrating Eq. (10) gives the relation between pressure at a reference altitude $h_0$ and any altitude h:,

$$p(h) = p(h_0) - \int_{h_0}^{h} \rho g dz \qquad (11)$$

Using Eq. (9), we can solve for $T(h)$

$$T(h) = \frac{p(h)}{\rho(h)R_g} = \frac{p(h_0)}{\rho(h)R_g} - \frac{1}{\rho(h)R_g}\int_{h_0}^{h} \rho g dz$$

$$= T(h_0)\frac{\rho(h_0)}{\rho(h)} - \frac{1}{\rho(h)R_g}\int_{h_0}^{h} \rho g dz \qquad (12)$$

Equation (12) is used to calculate the temperature in this study. $T(h_0)$ is the temperature at the reference altitude $h_0$. In this paper, the reference altitude of 80 km was used, the corresponding temperature was obtained from the model ECMWF.



### 3.4 Acceleration

Based on Newton's second law, the aerodynamic force can be expressed in terms of the aerodynamic acceleration $\boldsymbol{a}_a$. The acceleration of the FFUs are made up of a gravity acceleration and an aerodynamic acceleration. The resultant acceleration $\boldsymbol{a}_{ECEF}$ obtained from GPS solutions is the acceleration with respect to the ECEF frame. In order to obtain the inertial acceleration, the Coriolis and the centrifugal effects should be added in (Psiaki et al.),

$$\boldsymbol{a}_{IN/ECEF} = \boldsymbol{a}_{ECEF} + 2\boldsymbol{\omega}_E \times \boldsymbol{v}_{ECEF} + \boldsymbol{\omega}_E \times (\boldsymbol{\omega}_E \times \boldsymbol{r}_{ECEF}) \tag{13}$$

where the bold letters denote vectors, $\boldsymbol{a}_{IN/ECEF}$ is the inertial acceleration, $\boldsymbol{\omega}_E$ is the Earth rotational angular velocity (directed along the ECEF +Z axis), and $\boldsymbol{v}_{ECEF}$ and $\boldsymbol{r}_{ECEF}$ are the velocity and the position of the FFU in ECEF coordinates.

The gravitational acceleration vector $\boldsymbol{g}$ can be obtained from the law of universal gravitation together with the $J_2$ effect (Humi, 2007),

$$\boldsymbol{g} = -\frac{GM}{r^3}\left[1 - \frac{3}{2}J_2\left(\frac{R}{r}\right)^2\left(5\frac{z^2}{r^2} - 1\right)\right]\boldsymbol{r}$$

$$- [0,\, 0,\, 3J_2\left(\frac{R}{r}\right)^2 \frac{GMz}{r^3}]^{\mathrm{T}} \tag{14}$$

where $G$ is the universal gravitational constant, $M$ is the mass of the earth, $R$ is the mean radius of the Earth, $J_2 = 1.0826 \times 10^{-3}$, and $\boldsymbol{r} = [x, y, z]^{\mathrm{T}}$.

Therefore, the aerodynamic acceleration is

$$\boldsymbol{a}_a = \boldsymbol{a}_{IN/ECEF} - \boldsymbol{g}$$
$$= \boldsymbol{a}_{ECEF} + 2\boldsymbol{\omega}_E \times \boldsymbol{v}_{ECEF} + \boldsymbol{\omega}_E \times (\boldsymbol{\omega}_E \times \boldsymbol{r}_{ECEF}) - \boldsymbol{g} \tag{15}$$

### 3.5 Dynamic model

According to Newton's second law, the drag acceleration vector $\boldsymbol{a}_d = \boldsymbol{D}/m$, where $m$ denotes the mass of the FFU. Decomposing $\boldsymbol{a}_d$ in the local ENU frame, we have

$$\frac{AC_D\rho V}{2m}(v_x - w_x) - a_{dx} = 0 \tag{16a}$$

$$\frac{AC_D\rho V}{2m}(v_y - w_y) - a_{dy} = 0 \tag{16b}$$

$$\frac{AC_D\rho V}{2m}(v_z - w_z) - a_{dz} = 0 \tag{16c}$$

Assume that in the local ENU frame the vertical wind velocity is negligible $w_z = 0$. Hence, Eq. (16) is a set of 3 nonlinear equations with 3 unknowns, i.e., the unknown density and horizontal wind. By taking the division between equations, we can derive further

$$w_x = v_x - v_z \frac{a_{dx}}{a_{dz}} \tag{17a}$$





$$w_y = v_y - v_z \frac{a_{dy}}{a_{dz}} \tag{17b}$$

$$\rho = \frac{2m a_{dz}}{A C_D V v_z} \tag{17c}$$

Therefore, if the drag acceleration is known, the wind velocity can be obtained, and thus the density can be derived.

However, the horizontal wind is sensitive to the drag acceleration, since it is small compared to the sphere's velocity. Usually, the horizontal wind is no more than $150\,\mathrm{ms^{-1}}$ in the upper atmosphere (Faucher et al., 1963). As for the density calculation, the horizontal wind can be neglected (Faucher et al., 1963; Faucher and Morrissey, 1971; Salah, 1967). If Magnus lift is ignored, the aerodynamic acceleration vector $\boldsymbol{a}_a$ of the FFU is only the drag acceleration vector $\boldsymbol{a}_d = \boldsymbol{a}_a$, then 

$$\rho = \frac{2m a_{az}}{A C_D V v_z} \tag{18}$$

### 3.6 Iteration

In order to solve for a density in Eq. (18), a corresponding drag coefficient is needed. This drag coefficient can be interpolated according to Sec. 3.2, given an initial density. Figure 9 illustrates the algorithm. First, a starting profile of a density and a temperature from the ECMWF model as the standard atmospheric model are inserted. Then, a drag coefficient is obtained  through interpolation, so that a new density was obtained. The density as the new initial value equals the old density plus half of the difference of subtracting the old density from the new density, i.e. $\lambda = 0.5$ in Fig. 9, produces fast convergence. Afterwards, a corresponding temperature profile was computed via Eq. (12). Using Sec. 3.2, a new drag coefficient was calculated. In this way, the iteration continues until the relative change of two successive density iterations becomes smaller than $10^{-5}$.

### 4 Results

Figure 10 shows the calculated densities, compared to the values from the NRLMSISE-00 model, ECMWF and Radiosonde measurements. The density ratio is the division of the density by the one from the ECMWF model. In general, the obtained density profiles agree with the ECMWF curve below 70 km, with ratios between 0.87 and 1.07. The calculated density starts to approach the NRLMSISE-00 density, and to deflect from the ECMWF density above 70 km as the ratio between the calculated density and that from NRLMSISE-00 is around 1.2. This indicates that the calculated density is accurate, yet the accuracy is  somewhat lower above 70 km than below 70 km.

Figure 11 presents the temperatures from different data sources. The temperatures from LW2 and LW4 agree well generally, their maximum difference is around 3 K. They agree with the ECMWF temperature better below than above 50 km, with maximum differences around 10 K and 30 K, respectively. Below 26 km, the calculated temperatures have some oscillations. While some of the altitude dependence is recognizable in the radiosonde data, the largest changes are between 23 and 27 km,  not seen in R data.





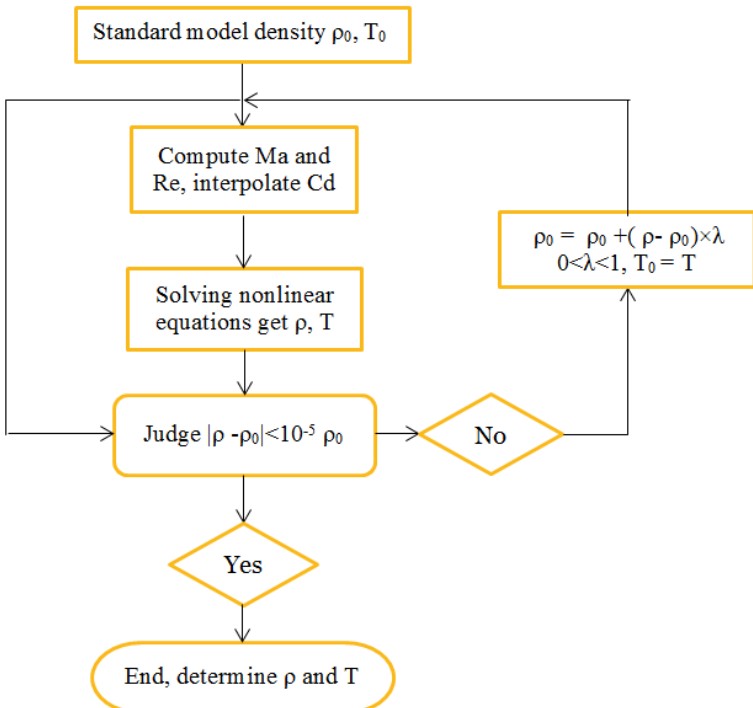

**Figure 9.** Algorithm flow chart

Figure 12 shows wind speeds below 80 km, the wind speeds from measurements of the two spheres cover 11–80 km, based on Eq. (17), neglecting the Magnus lift. In the E/W direction, the wind speeds from the two spheres agree well with differences smaller than $1\,\mathrm{ms^{-1}}$ below 45 km, agree with a maximum difference of $7\,\mathrm{ms^{-1}}$ between 45 and 62 km, and have discrepancies in between $7$–$23\,\mathrm{ms^{-1}}$ above 62 km. In the N/S direction, below 40 km, the differences are smaller than $1\,\mathrm{ms^{-1}}$, between 40 and 55 km, they are smaller than $5\,\mathrm{ms^{-1}}$. At most altitudes between 55 and 78 km, the differences are in the range $10$–$32\,\mathrm{ms^{-1}}$, above 78 km, they increase rapidly from 32 to $55\,\mathrm{ms^{-1}}$. As for the Radiosonde measurements, the calculated speeds match them well with a maximum difference around $10\,\mathrm{ms^{-1}}$, which only cover altitudes below 33 km. Regarding to the ECMWF values, in the E/W direction, the calculated speeds have large differences from them, the differences are larger than $20\,\mathrm{ms^{-1}}$ at most altitudes, and reach $70\,\mathrm{ms^{-1}}$ quickly around 80 km. In the N/S direction, the calculated speeds differ from the ECMWF values more than $20\,\mathrm{ms^{-1}}$ at most altitudes above 40 km, and the differences increase to $125\,\mathrm{ms^{-1}}$ rapidly. In summary, the calculated wind speeds make sense with acceptable uncertainties below 45 km.





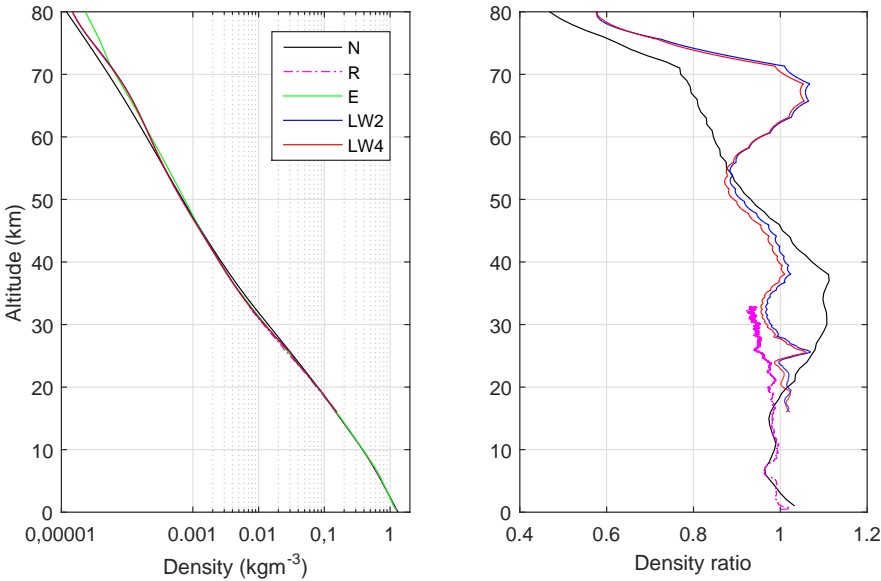

**Figure 10.** Comparison of density from NRLMSISE-00 (N), Radiosonde (R), ECMWF at Esrange (E) as well as the calculated densities. (a) is the absolute density value, (b) is the ratio of the density with respect to E.

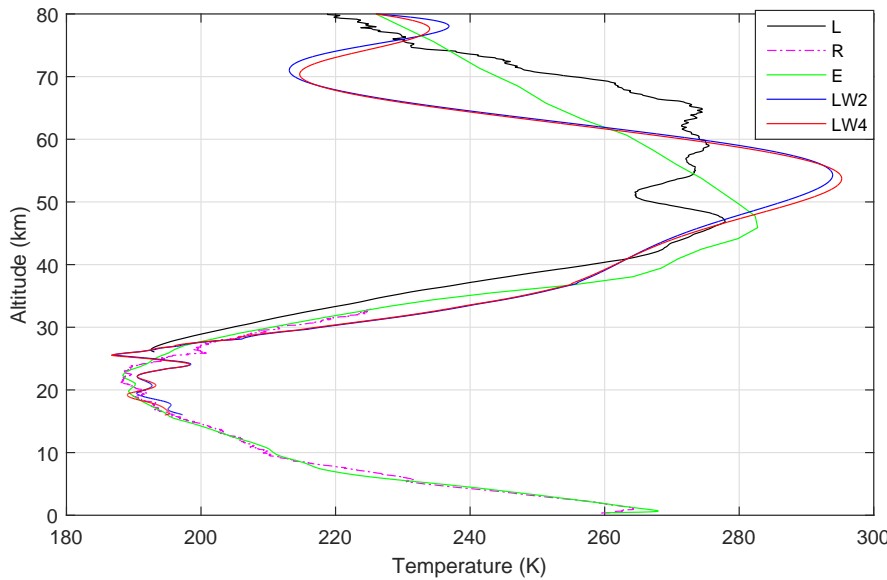

**Figure 11.** Comparison of temperature from Lidar (L), Radiosonde (R), ECMWF at Esrange (E) as well as the calculated temperature





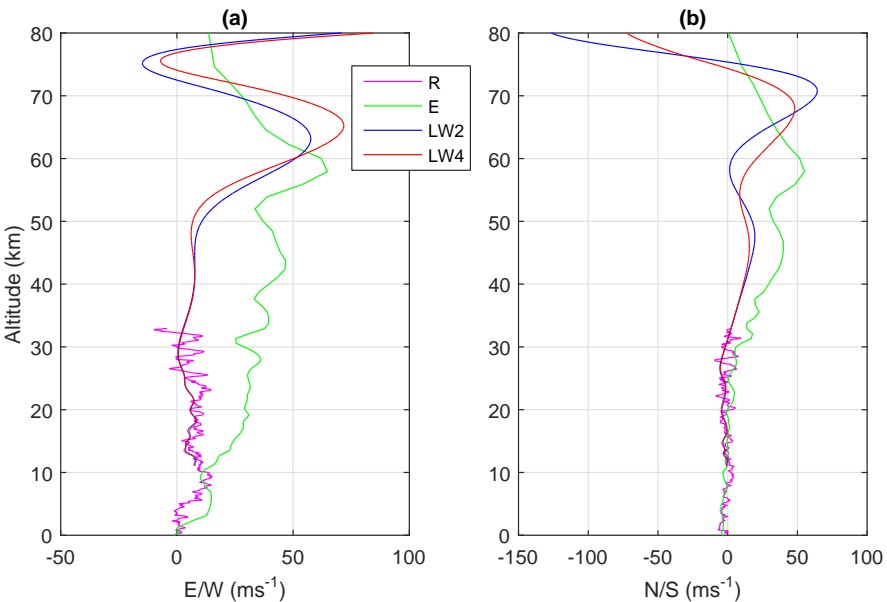

**Figure 12.** Wind speed comparison from the closest Radiosonde data to the launch time (R), ECMWF at apogee (E) and the calculated wind speeds

## 5   Discussion

From Eq. (18), the density is affected by the total acceleration from numeric differentiation of the GPS velocity, the velocity of the sphere relative to the air flow, the drag coefficient and other factors such as mass and cross-section area. The mass and the size of the sphere are available with rather high precision, and would only introduce a systematic bias to the density.

The accuracy of the acceleration is influenced by GPS measurements and processing, and estimation of the acceleration. In the case of small and rigid sphere the rotational motion becomes important (as compared to for example large inflatable sphere), especially in configurations when the sphere is ejected from a spinning rocket. From Fig. 2 it is apparent that with the presented technique, getting acceleration accuracy below $0.1~\mathrm{ms^{-2}}$ is challenging, as illustrated by irregular dependence of both the value of the drag acceleration and its angle to the velocity of the sphere before 230 s. Part of this inaccuracy seems to be related to filtering of the oscillation. A solution with full attitude reconstruction of the sphere would allow compensation for some of this inaccuracy by correcting for the displacement of the antenna phase centre and the centre of mass. This would relieve the need of filtering, thus improving the altitude resolution as well. Some challenges would still remain, related to the variation of the position of the phase centre of the antenna for various viewing angles to the satellites.

The density determination is also affected by the drag coefficient. while the dependence of the drag coefficient on the flow parameters is well behaved in the supersonic regime (in our case, corresponding to altitudes above 30 km), it exhibits fast variations in the dense, high Re subsonic flows. This is seen both as the sharp decrease from almost 1 to 0.5 over the range





of just a few km (see Fig. 7(d)), and the irregular variations around 0.5 for altitudes below 25 km. This sensitivity of the drag coefficients on the parameters introduces systematic errors, which might be responsible for the positive deviation of the density 320 ratio at 25–28 km (see Fig. 10), and lead to temperature oscillations in this area.

Temperature estimates agree reasonably well with independent data below 50 km, while exhibit larger variations above that. This is due to less reliable density reconstruction. Also, the systematic effect of the reference altitude temperature taken from the model is the highest at the top altitudes, becoming negligibly lower down in the atmosphere.

The wind speed is affected by the drag acceleration, according to Eq. (17), while the drag acceleration depends on the 325 Magnus lift in the aerodynamic force. The Magnus lift is also related to the attitude motion, and must be accounted for to determine the wind speed with high precision.

# 6   Conclusions

This paper presents the active falling sphere technique using a low-cost GPS receiver to measure atmospheric densities, temperatures, and horizontal wind speeds. Densities, temperatures, and horizontal wind speeds below 80 km were obtained, and they 330 compare well with the independent data. As the altitude goes down, the accuracy becomes higher. The densities have relative differences within 10% with respect to the ECMWF reference values below 70 km, while the temperatures have absolute differences smaller than 10 K relative to this reference model below 50 km. The wind speeds from two spheres coincide very well below 45 km, with estimated uncertainties of $20\,\mathrm{ms}^{-1}$. In order to make the calculated densities valid, the sphere's velocities and accelerations should be large enough, for example, the Mach number of the velocity is greater than 3 above 40 km, the 335 drag acceleration is greater than $1\,\mathrm{ms}^{-2}$. This is to reduce the effect of the wind speed and the acceleration uncertainty. To make the results more precise, the attitude motion and the resulting Magnus lift must be included, since the sphere rotated over the flight time.

*Acknowledgements.*  Y. Yuan is grateful for PhD studentship financial support from the Chinese Scholarship Council.

The LEEWAVES project was funded by Swedish National Space Board (SNSB). 340



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
