# Peer review of "Atmosphere Density Measurements Using GPS Data from Rigid Falling Spheres"

_Atmospheric Measurement Techniques, 2017_

## Referee Comment (RC1) · Anonymous Referee #1 · 29 May 2017

Major concerns ——————-

An error calculation from known uncertainties like trajectory uncertainty or smoothing length is missing and needed to evaluate the precision of the new method. The current presented comparison to ECMWF model data allows only hand-waving estimation of the precision of the measurements.

The authors speculate about the importance of the Magnus effect on the results of their measurement throughout the whole manuscript about 10 times. A simple estimation of the magnitude of the effect is needed. Maybe the rocket spin period or the modulation period of the GPS positions would allow to estimate limits for the Magnus effects.

Most figure captions do not give sufficient information. Only the caption to figure 6 gives sufficient information. Please expand the captions so the reader can understand

what is plotted without guessing. The information is often given in the text somewhere, but there it distracts the reader and makes understanding of the figures difficult. One example: The caption of figure 2 should at least hold the definition of the angle and the residuals.

In the results section the authors estimate the precision of their measurements by comparison of the rigid sphere measurements to the ECMWF model in the middle atmosphere. The discussion is mostly hand waving and lacks careful discussion. Statements like "This indicates that the calculated density is accurate, yet the accuracy is somewhat lower above 70 km than below 70 km." (line 285) are not convincing. This approach of estimating the precision seems inappropriate. This approach seems wrong, especially since the authors state in the manuscript on line 129: "Whereas the densities from ECMWF are reliable below 50 km, as the altitude goes up, the uncertainties increase"

Minor comments ――――――――

The abstract is not well written. It is to short and not descriptive of the actual manuscript. It is partly just a list of basic statements that were not developed in the manuscript. One example for such a basic statement is: "Aerodynamic drag relates atmospheric densities to other variables such as velocities of spheres, drag coefficients, and reference area. "

The abstract does not give essential information like where, when, and with what vehicle the measurements were performed. It would be valuable to add information about the vertical resolution to achieve a defined precision limit as well as the altitude range where this is achieved.

Section 2.2 should be revised. There are paragraphs that consist of single sentences.

Line 16: Instead of "Li et al., 2013" it should read "e.g. Li et al., 2013"

Line 31: Did Martineau, 2012 actually measured winds or did they only used simulated

data?

Line 66: Please clarify why these two FFUs were selected.

Line 74: Please clarify if the ENU is fixed during the flight or varying with time.

Line 81: The enhanced accelerations in x and y need to be discussed. Otherwise the enhanced accelerations might be misinterpreted as a problem in calculating the ENU.

Fig.1: The x-axis scaling of subfigure c is different than the others

Line 99: "the acceleration estimates are valid below 70–80 km." This statement seems hand waving and needs to be expanded. What is the precision threshold that defines "valid"? Is the upper limit 70 or 80 km? Figure 2 does not show altitudes but the discussion of Fig. 2 would benefit from having an altitude scale.

Line 124: What is meant by "NRLMSISE-00 generated the data"?

Line 125: What is the actual activity level used?

Line 126: "at the apogee" should likely be "below apogee" or something more descriptive.

Line 128: Please provide reference for this statement. Le Pichon et al. discuss only temperatures and winds. Le Pichon et al. state "... consitent up to ∼40 km". Please clarify where the limit of 50 km comes from.

Figure 3: Please mark the altitude levels of the ECMWF data. The ECMWF data seems to be interpolated, please specify what interpolation was used.

Figure 5: Judging from the difference of E2 and E3 and the location of the polar vortex (Fig 6) it seems likely that data E1 and E2,E3 are not consistent. Please check the data again.

Line 201: "kgs" space or dot missing between kg and s.

Line 222: "... will not be in hydrostatic equilibrium". The following discussion assumes

hydrostatic equilibrium. Please clarify.

Figure 8: It will be useful to draw the trajectory of the actual Re and Ma numbers of the flight. The figure caption should hold the source of the data.

Line 239: "Pasiaki et al., )": Year missing

Line 265: What numerical method is used to calculate these derivates?

Line 267: The references according the windspeed in the upper atmosphere seems outdated. There are many more wind obsevations that show wind measurements in the uppper atmosphere since 1963. Maybe taking a upper limit from HWM-14 would be more up to date: http://onlinelibrary.wiley.com/doi/10.1002/2014EA000089/full

Line 279: From Fig. 9 and the text it is unclear how different altitudes are treated. Please clarify, if the relative change is the absolute sum of all residuals.

Figure 10: It is likely that some of the small scale structures in the density ratio is caused by interpolation of the different dataset on a uniform grid. Infomation is needed to clarify this.

What is the cause of the enhanced density from LW2 and LW4 around 25 km.

Figure 11: The temperatures from LW2 and LW4 look very smooth. This indicates that dependend data points were drawn. Please provide information about the actual measurement resolution and the smoothing length used to calculate density and temperature from the actual acceleration profiles. It would be helpful to plot the actual measurements grid.

Line 310: Where does this threshold value come from. What precision of density, temperature and wind does it correspond to?

---

## Referee Comment (RC2) · Anonymous Referee #2 · 13 Jul 2017

General comments:

The authors present the results from the first analysis of two active falling sphere measurements covering altitudes from approximately 80 km to the surface. Although falling spheres have been used extensively in the past to obtain density, temperature, and wind measurements, the instrumented rigid falling spheres used by the authors represent a new development in the technique with potential improvements in the measurements.

The description of the instrumentation is adequate, although not very detailed for a measurement techniques article. The authors provide an excellent literature review of similar measurements made in the past, going back to the early days of aeronomy and space studies.

[Figure]

Overall, the technique and the description of the measurements are interesting. My concern, however, is the general characterization that the results are in good agreement or general agreement with the independent measurements or model estimates for the same time and location. There is clearly good agreement below 20 km, but presumably, that is not the altitude range where this technique will be most valuable. Above 30 km, there are very large differences between the falling sphere estimates and the other profiles. The density ratios are the most difficult to assess since the only basis for comparison are the values from a dynamical model (ECMWF) and an empirical model (NRLMSIS). Nonetheless, difference ratios exceeding 10% would concern me and would certainly be expected to affect the dynamics significantly if the measurements are used for that type of analysis. The lidar data actually provide a temperature measurement that can be compared directly to the falling sphere values, and again, significant differences of 30K or more are found. The wind estimates are difficult to assess since only model comparisons are available above 30 km. The model winds at a specific location are probably the least accurate of the parameters used in the comparisons.

The description of the analysis technique, the first results, and the available comparisons should be published, but the results should be characterized more realistically. The results from the first analysis presented here do not compare particularly well with the available independent measurements and model estimates, but they indicate the potential of the technique. The authors suggest improvements in the analysis technique that can be applied to refine the results. Perhaps those refinements can produce better agreement and should be emphasized in the discussion.

Specific comments:

lines 213-214. "...as it can provide data up to 150 km." The statement appears to refer to MSIS, but the empirical model provides values to altitudes higher than 150 km. Please clarify.

lines 218-219. Why is there a cut-off for the drag coefficient at 95 km? The density and temperature values used to estimate the drag coefficient appear to cover a broader range of altitudes. Please clarify.

lines 234-235. Why was 80 km chosen as the reference altitude? The subsequent analsysis is limited to the height range below 80 km, and that is presumably the reason for the choice, but why were the higher altitudes ignored? Please explain.

line 243. Explain the J2 effect briefly to make the paper more self-contained.

line 258. The assumption of zero vertical wind is a practical choice that most likely affects the density estimate most directly. Can the authors estimate the magnitude of the potential error introduced by this assumption? The vertical winds in the mesosphere can be large, of the order of several meters per second.

lines 282-285. Ratios of 0.87 to 1.07 for the density do not represent particularly good agreement. Objectively, this could represent the difference between cyclonic and anticyclonic flow, for example. "This indicates that the calculated density is accurate..." Quite the contrary seems to be the case.

lines 287-289. Similar comments apply to the discussion of the temperature comparisons, although the comparison with the lidar measurements make the differences even more problematical since the lidar data represent an actual measurement rather than a model estimate.

lines 330-331. Is the conclusion warranted? If such large differences are acceptable, what is the objective basis for making that determination?

Some of the differences between the falling sphere values and the independent measurements and model estimates could be due to geophysical variations rather than instrumental error. Discussion of such effects would be helpful.
* * *

---

## Author Comment (AC1) · 10 Aug 2017

Answers to comments by reviewer 1

Major concerns

1. An error calculation from known uncertainties like trajectory uncertainty or smoothing length is missing and needed to evaluate the precision of the new method. The current presented comparison to ECMWF model data allows only hand-waving estimation of the precision of the measurements.

Thank you for your comment. Now the precision analysis for the density is done, which is added in Discussion Section.

2. The authors speculate about the importance of the Magnus effect on the results of their measurement throughout the whole manuscript about 10 times. A simple estimation of the magnitude of the effect is needed. Maybe the rocket spin period or the modulation period of the GPS positions would allow to estimate limits for the Magnus effects.

The Magnus effect is estimated in Section 3.1 Aerodynamic force, the ratio between the Magnus lift and the drag is around 0.004 in the extreme case, thus it can be neglected. The reference to Magnus effect is now minimized, but we still include the reference to it and the estimate.

3. Most figure captions do not give sufficient information. Only the caption to figure 6 gives sufficient information. Please expand the captions so the reader can understand what is plotted without guessing. The information is often given in the text somewhere, but there it distracts the reader and makes understanding of the figures difficult. One example: The caption of figure 2 should at least hold the definition of the angle and the residuals.

Captions are changed to make it clear.

4. In the results section the authors estimate the precision of their measurements by comparison of the rigid sphere measurements to the ECMWF model in the middle atmosphere. The discussion is mostly hand waving and lacks careful discussion. Statements like "This indicates that the calculated density is accurate, yet the accuracy is somewhat lower above 70 km than below 70 km." (line 285) are not convincing. This approach of estimating the precision seems inappropriate. This approach seems wrong, especially since the authors state in the manuscript on line 129: "Whereas the densities from ECMWF are reliable below 50 km, as the altitude goes up, the uncertainties increase"

Thanks for pointing out this, the confusing text is changed.

Between 50 and 70 km, the ratio between the calculated density and the one from ECMWF is between 0.87 and 1.06, however the ECMWF value is not reliable, the accuracy must be analysed in another way, which is conducted in Sec. 5. Here, we can see that the calculated density is reliable below 50 km and above 70 km. If we trust the reference density completely, we can summarize that the accuracy is somewhat lower above 70 km than below 50 km.

Of course, there is a possibility of the geophysical difference between the densities observed in a point measurement from a model assimilating scarce measurement points.

Minor comments

1. The abstract is not well written. It is to short and not descriptive of the actual manuscript. It is partly just a list of basic statements that were not developed in the manuscript. One example for such a basic statement is: "Aerodynamic drag relates atmospheric densities to other variables such as velocities of spheres, drag coefficients, and reference area. "

The abstract does not give essential information like where, when, and with what vehicle the measurements were performed. It would be valuable to add information about the vertical resolution to achieve a defined precision limit as well as the altitude range where this is achieved.

The abstract is modified according to your comment.

2. Section 2.2 should be revised. There are paragraphs that consist of single sentences.

Sectoin 2.2 is revised now.

3. Line 16: Instead of "Li et al., 2013" it should read "e.g. Li et al., 2013"

The reference is fixed

4. Line 31: Did Martineau, 2012 actually measured winds or did they only used simulated data?

In fact, this thesis is a very advanced piece of work. It refers to an actual experiment, launched on a sounding rocket. Unfortunately, there are no published results from this experiment. There are a couple of AGU general assembly talks, but the only detailed publication is the thesis by Martineau.

From his thesis, they took measurements by employing rigid falling spheres, he presented the mathematical model to derive the density, but did not show the model for the winds or analyze data. In fact, there were issues with the precision accelerometers saturating due to the sphere rotation.

5. Line 66: Please clarify why these two FFUs were selected.
These two FFUs had the best quality data to be analyzed. Another FFU had data of considerably inferior quality, while the GPS raw data from the fourth FFU was lost.

6. Line 74: Please clarify if the ENU is fixed during the flight or varying with time.

It is a local reference frame varying with time, to show wind speeds visually.

7. Line 81: The enhanced accelerations in x and y need to be discussed. Otherwise the enhanced

accelerations might be misinterpreted as a problem in calculating the ENU.

Do you mean the accelerations shown in Fig.1(c)? They are resultant accelerations, which are added in the figure caption and the text now.

8. Fig.1: The x-axis scaling of subfigure c is different than the others

Fixed

9. Line 99: "the acceleration estimates are valid below 70–80 km." This statement seems hand waving and needs to be expanded. What is the precision threshold that defines "valid"? Is the upper limit 70 or 80 km? Figure 2 does not show altitudes but the discussion of Fig. 2 would benefit from having an altitude scale.

Estimating uncertainties of aerodynamic accelerations are smaller than 0.1 $m/s^2$, and considering the aerodynamic accelerations are around 0.3 and 1 $m/s^2$ at 80 and 72 km, the aerodynamic acceleration is valid at 80 km, and becomes more precise as the altitude goes down, since the aerodynamic acceleration becomes larger.

The altitude scale is from Fig. 1(a) that is added in the text now.

10. Line 124: What is meant by "NRLMSISE-00 generated the data"?
NRLMSISE-00 is an empirical model, which can output the atmospheric data at assigned altitudes.

11. Line 125: What is the actual activity level used?
The magnetic AP index (7 different time intervals), and the F10.7 flux (2 different time intervals) were entered when we run NRLMSISE-00, although the effects of AP and F10.7 are neither large nor well established below 80 km. The indices are taken from the National Geophysical Data Center (ftp://ftp.ngdc.noaa.gov/STP/GEOMAGNETIC_DATA/INDICES/KP_AP).
The magnetic AP index input consists of:
1) the daily magnetic index (5),
2) 3 hour AP for current time (12),
3) 3 hour AP for 3 hours before current time (7),
4) 3 hour AP for 6 hours before current time (7),
5) 3 hour AP for 9 hours before current time (5),
6) average of eight 3 hour AP indices from 12 to 33 hours prior to current time (1.375), and,
7) the average of eight 3 hour AP indices from 36 to 57 hours prior to current time (11.75).

The F10.7 flux input consists of:
1) the daily average for the previous day (97.3), and
2) the 81 day average centred on the day of interest (100.47).

12. Line 126: "at the apogee" should likely be "below apogee" or something more descriptive.

*It refers to the longitude and the latitude of the apogee. The whole trajectory can be projected on the surface of the Earth in terms of longitude and latitude, and we use the coordinates of the apogee point here.*

13. Line 128: Please provide reference for this statement. Le Pichon et al. discuss only temperatures and winds. Le Pichon et al. state "... consitent up to ~40 km". Please clarify where the limit of 50 km comes from.

*Thank you for your pointing this! The limit should be 40 km instead of 50 km. The related portions of the text are changed.*

14. Figure 3: Please mark the altitude levels of the ECMWF data. The ECMWF data seems to be interpolated, please specify what interpolation was used.

*The ECMWF data are denoted by '.' now. The altitude resolution of ECMWF changes between 20 and 4400 m. The altitude resolution of NRLMSISE-00 is 1000 m. The linear interpolation of the NRLMSISE-00 altitudes on the ECMWF data is used.*

15. Figure 5: Judging from the difference of E2 and E3 and the location of the polar vortex (Fig 6) it seems likely that data E1 and E2, E3 are not consistent. Please check the data again.

*The data have been checked again, and no problems were discovered. The difference is because the data do not cover the same longitude and latitude. Since the trajectory moved across different longitudes and latitudes, we think it is good to show some representing locations.*

16. Line 201: "kgs" space or dot missing between kg and s.

*Fixed*

17. Line 222: "... will not be in hydrostatic equilibrium". The following discussion assumes hydrostatic equilibrium. Please clarify.

*Thanks for your question. It is a typo, 'not' should be removed.*

18. Figure 8: It will be useful to draw the trajectory of the actual Re and Ma numbers of the flight. The figure caption should hold the source of the data.
*Fixed*

19. Line 239: "Pasiaki et al., )": Year missing

*Fixed*

20. Line 265: What numerical method is used to calculate these derivates?

A finite difference of the velocities is used to calculate the acceleration.

21. Line 267: The references according the wind speed in the upper atmosphere seems outdated. There are many more wind obsevations that show wind measurements in the uppper atmosphere since 1963. Maybe taking a upper limit from HWM-14 would be more up to date: http://onlinelibrary.wiley.com/doi/10.1002/2014EA000089/full

Thank you, this is now fixed.

22. Line 279: From Fig. 9 and the text it is unclear how different altitudes are treated. Please clarify, if the relative change is the absolute sum of all residuals.

To make it clear, two sentences are added.

The iteration is performed at one single altitude each time. When a density and a temperature are determined at the current altitude, the iteration moves to next altitude.

23. Figure 10: It is likely that some of the small scale structures in the density ratio is caused by interpolation of the different dataset on a uniform grid. Information is needed to clarify this.

You are right. The interpolation causes the small scale structures.

24. What is the cause of the enhanced density from LW2 and LW4 around 25 km.

This is discussed in the discussion section.

This sensitivity of the drag coefficients on the parameters introduces systematic errors, which might be responsible for the positive deviation of the density ratio at 25−28 km (see Fig. 10)

25. Figure 11: The temperatures from LW2 and LW4 look very smooth. This indicates that dependent data points were drawn. Please provide information about the actual measurement resolution and the smoothing length used to calculate density and temperature from the actual acceleration profiles. It would be helpful to plot the actual measurements grid.

The altitude resolution is between 1 and 12 m, and the actual measurement grids are plotted, but it is not distinguishable.

26. Line 310: Where does this threshold value come from? What precision of density, temperature and wind does it correspond to?

The threshold value come from fig.2(b) and 2(c). First, at apogee, the aerodynamic acceleration should be very low. Second, the aerodynamic acceleration should be opposite to the velocity.

In general, with the altitude going down, the relative uncertainty of the aerodynamic acceleration

becomes smaller, the precision of density becomes higher. The corresponding precisions are 30%, 5%, 2%, 1% at 80, 70, 60, 54 km. The temperature precision should be not affected, according to the reference: http://onlinelibrary.wiley.com/doi/10.1029/91JD02395/full. The wind precision depends, if the acceleration uncertainty is in the same direction of the acceleration, the wind precision will not affected.

---

## Author Comment (AC2) · 10 Aug 2017

Answers to reviewer 2

1. lines 213-214. "...as it can provide data up to 150 km." The statement appears to refer to MSIS, but the empirical model provides values to altitudes higher than 150 km. Please clarify.

Indeed, MSIS can provide values to altitudes higher than 150 km. Since our GPS data only extend to the apogee around 138 km, 150 km is mentioned in the paper. To make it clear, 150 km is removed in the paper.

2. lines 218-219. Why is there a cut-off for the drag coefficient at 95 km? The density and temperature values used to estimate the drag coefficient appear to cover a broader range of altitudes. Please clarify.

The drag coefficients come from experiment data of Bailey and Hiatt (1972). They are only available for certain ranges of Mach and Reynolds numbers, which correspond to an altitude range between 16 and 95 km in the paper.

3. lines 234-235. Why was 80 km chosen as the reference altitude? The subsequent analsysis is limited to the height range below 80 km, and that is presumably the reason for the choice, but why were the higher altitudes ignored? Please explain.

80 km is chosen according to the valid acceleration data. Please look at Fig. 2(b-c), 2(b) is the value of the aerodynamic acceleration, 2(c) is the angle between the aerodynamic acceleration and the velocity. As we know that at apogee around 115 s, the aerodynamic acceleration should be very small, but it is around 0.1 m/s$^2$ in Fig. 2(b) for LW2, thus we estimate conservatively that the uncertainty of the aerodynamic acceleration is 0.1 m/s$^2$. Above 80 km (about 224 s), the accelerations are very small, around 0.1 m/s$^2$ for LW2, and even smaller for LW4. In addition, the angle between the aerodynamic acceleration and the velocity should be around 180 degrees, but it is smaller than 150 deg above 80 km for LW2. Hence, the data higher than 80 km are not valid, therefore not presented.

To try to recover the information from higher altitudes, a complete reconstruction of attitude of the FFUs should be done, which is outside the scope of the first analysis of the data presented here.

4. line 243. Explain the J2 effect briefly to make the paper more self-contained.

Since the Earth is not a perfect sphere, the irregular shape of the Earth makes the gravitational field is not exactly central. The most important correction term is J2 term. This is now explained in the paper in a better way.

5. line 258. The assumption of zero vertical wind is a practical choice that most likely affects the density estimate most directly. Can the authors estimate the magnitude of the

potential error introduced by this assumption? The vertical winds in the mesosphere can be large, of the order of several meters per second.

Assume that the vertical wind is smaller than 1 m/s below 50 km, and smaller than 5 m/s between 50 and 80 km. From Fig. 1, below 50 km, the vertical velocity decreases from larger than 1000 m/s to about 200 m/s, between 50 and 80 km, it is larger than 1000 m/s. Hence, the relative uncertainty of the vertical velocity is smaller than 0.5%, as shown in the discussion part of the new version, which is smaller than uncertainties of the aerodynamic acceleration and the drag coefficient, thereby having only a minor effect on the density. The error transfer is discussed in the new version.

6.  lines 282-285. Ratios of 0.87 to 1.07 for the density do not represent particularly good agreement. Objectively, this could represent the difference between cyclonic and anticyclonic flow, for example. "This indicates that the calculated density is accurate..." Quite the contrary seems to be the case.

It is a challenge to evaluate the calculated density exactly. The ratio is with respect to the ECMWF value, however, this model is has some uncertainties. In the new version, the error transfer is analyzed in the discussion part. The uncertainties of the densities in the extreme cases are 37%, 9.3%, 5.5%, and 4.5% respectively, at 80, 70, 60, 54 km.

7.  lines 287-289. Similar comments apply to the discussion of the temperature comparisons, although the comparison with the lidar measurements make the differences even more problematical since the lidar data represent an actual measurement rather than a model estimate.

It is also challenging to evaluate the temperature exactly. It is good to have lidar measurements. Unfortunately, because of tropospheric cloudiness, lidar information was not available concurrent with the LEEWAVES launch. Instead, comparisons in this paper are based on lidar measurements obtained during the day prior to the launch. The temperature precision of a falling sphere are not affected, according to (Schmidlin et al., 1991), the density bias does not affect the temperature.

8.  lines 330-331. Is the conclusion warranted? If such large differences are acceptable, what is the objective basis for making that determination?

The formulation is changed in the paper.

9.  Some of the differences between the falling sphere values and the independent measurements and model estimates could be due to geophysical variations rather than instrumental error. Discussion of such effects would be helpful.

That makes sense, since the independent measurements and the model estimates are not exact at the same moments or in the same areas as the measurements of the falling spheres. We

extended the discussion in the paper now.